# ModernBERT + ColBERT: Enhancing biomedical RAG through an advanced re-ranking retriever

## Abstract

Retrieval-Augmented Generation (RAG) is a powerful technique for enriching Large Language Models (LLMs) with external knowledge, allowing for factually grounded responses, a critical requirement in high-stakes domains such as healthcare. However, the efficacy of RAG systems is fundamentally restricted by the performance of their retrieval module, since irrelevant or semantically misaligned documents directly compromise the accuracy of the final generated response. General-purpose dense retrievers can struggle with the nuanced language of specialised domains, while the high accuracy of in-domain models is often achieved at prohibitive computational costs. In this work, we aim to address this trade-off by developing and evaluating a two-stage retrieval architecture that combines a lightweight ModernBERT bidirectional encoder for efficient initial candidate retrieval with a ColBERTv2 late-interaction model for fine-grained re-ranking. We conduct comprehensive evaluations of our retriever module performance and RAG system performance in the biomedical context, fine-tuning the IR module using 10k question-passage pairs from PubMedQA. Our analysis of the retriever module confirmed the positive impact of the ColBERT re-ranker, which improved Recall@3 by up to 4.2 percentage points compared to its retrieve-only counterpart. When integrated into the biomedical RAG, our IR module leads to a state-of-the-art average accuracy of 0.4448 on the five tasks of the MIRAGE question-answering benchmark, outperforming strong baselines such as MedCPT (0.4436). Our ablation studies reveal that this performance is critically dependent on a joint fine-tuning process that aligns the retriever and re-ranker; otherwise, the re-ranker might degrade the performance. Furthermore, our parameter-efficient system achieves this result with an indexing speed over 7.5 times faster than leading baselines, providing a practical pathway for developing trustworthy biomedical RAG systems. Our implementation is available at: https://anonymous.4open.science/r/biorag-MC-9F3D/

## 1 Introduction

Retrieval-Augmented Generation (RAG) ((Lewis et al., 2020)) has become a crucial technique for enhancing the capabilities of Large Language Models (LLMs), particularly on domain-specific and knowledge-intensive tasks that require factually grounded responses. By incorporating up-to-date external knowledge, RAG enriches the LLM's context, enabling it to access information beyond its original training data and allowing for the generation of more accurate, fact-based responses, while reducing hallucinations (Fan et al. (2024)).

In fields such as healthcare, which demand factual accuracy and evidence-based responses, RAG is particularly promising. Training a foundational LLM for such a specific domain requires vast amounts of labelled data, which are often difficult and expensive to obtain. Furthermore, the clinical application of LLMs is highly susceptible to performance degradation due to distribution shift, a common phenomenon in medicine resulting from factors such as emerging pathogens, new therapeutic guidelines, and evolving standards of care (Peng et al. (2023), Ng et al. (2025), Tian et al. (2023)). An LLM without a mechanism to quickly integrate a massive amount of new information may provide factually incorrect responses with high confidence. RAG addresses these issues by

enabling the LLM to access real-time, domain-aligned information directly from the deployment environment (Fan et al. (2024)). This mechanism significantly reduces the probability of generating non-factual responses, ensuring that the output is grounded in verifiable, up-to-date data (Finardi et al. (2024), Xiong et al. (2024)).

However, RAG performance is intrinsically tied to the efficacy and effectiveness of the process of retrieving data from the external knowledge base. If the retrieved documents are not semantically aligned with the original query, the generative model is unlikely to benefit from this context to generate accurate responses (Hu & Lu (2025), Shi et al. (2023)). Although classical retrieval methods like BM25 (Sawarkar et al. (2024)) and DPR (Karpukhin et al. (2020)) conclusively demonstrated that RAG could increase the factual accuracy of LLM-generated responses, they occasionally struggle with the deep semantic understanding of specialised domains, which hinders their performance in settings such as healthcare, where language is heterogeneous and often ambiguous. For instance, lexical models may fail to recognise "Acute Myocardial Infarction" and "Heart attack" as the same condition.

To address this limitation and develop a more reliable and efficient medical RAG system, we propose a two-stage hybrid retrieval module featuring ModernBERT (Warner et al. (2024)) and ColBERT (Santhanam et al. (2021)). As the initial stage, we utilise a state-of-the-art ModernBERT bi-encoder for its speed and its ability to capture broad context during initial retrieval. Subsequently, for the re-ranking stage, we leverage ColBERT's late-interaction mechanism, which enables precise semantic alignment at the token level. We performed exhaustive experiments using the MIRAGE benchmark, with both models fine-tuned on a 20,000 subset of questions and passages from the widely known biomedical corpora PubMed and PubMedQA. Hence, this work presents:

- High-performance two-stage retrieval architecture. We empirically demonstrate that our ModernBERT + ColBERT pipeline achieves the highest average accuracy (0.4448) on the MIRAGE benchmark, outperforming leading models such as MedCPT (0.4436) and DPR (0.4174). Furthermore, our framework is established as the best in the MedMCQA task (0.4172).

- Balanced performance and efficiency. We show that our system achieves this high performance with outstanding efficiency. Our lightweight 149M-parameter ModerBERT model, trained on 20k data points, manages to index the knowledge base 7.5 times faster than the 220M-parameter MedCPT model, which was pre-trained on a massive 255M-pair dataset, establishing a highly efficient standard for clinical RAG.

- Empirical Proof of the "Fine-Tuning Requirement": We provide critical evidence that a deliberate, end-to-end fine-tuning strategy is crucial for achieving an effective and coordinated operation between the retriever and re-ranker, as a naive composition can actually degrade results.

## 2 RELATED WORK

Since its introduction by Lewis et al. (2020), Retrieval-Augmented Generation (RAG) has proven to be an effective technique for mitigating the inherent limitations of LLMs by extending their static parametric memory with external knowledge. This paradigm has evolved in the broader field of context engineering (Mei et al. (2025)), which seeks to optimise the processing and handling of context through diverse functions to maximise the expected quality of an LLM's response. For example, Knowledge Graph-Enhanced RAG aims to improve the context by moving from a static external knowledge base to one based on relational graphs, in order to provide richer information beyond what is contained in the unstructured text alone (Wu et al. (2024), Hu et al. (2025), Han et al. (2025)). Agentic RAG is an alternative approach to creating an optimal context for a given query; in this case, intelligent planning, implemented through an agent, is used to select and iteratively refine the most appropriate retrieval strategies to form the context (Singh et al. (2025)). However, the efficacy of any RAG framework, whether classic, graph-based or agentic, fundamentally depends on the quality of its underlying retrieval module. The medical sciences are particularly challenging in this regard due to the complexity and ambiguity of clinical language that hinders precise semantic capture, the sensitivity of the data, which limits training, and the high precision required in the generated responses (Ng et al. (2025)).

## 2.1 CLINICAL INFORMATION RETRIEVAL

Information Retrieval (IR) in the clinical field faces two primary challenges related to its rich and diverse vocabulary, as described by Tamine & Goeuriot (2021). The first is the lexical gap, which refers to the severe vocabulary mismatch between the query and the relevant passages. For instance, a query may be expressed in everyday terms (e.g., "stroke"), while the scientific corpus uses precise technical terminology (e.g., "cerebrovascular accident - CVA"). The second challenge is the semantic gap, which denotes the difference between the literal terms used in the text and the concepts they express. This issue arises from the use of different words or acronyms for the same symptom/treatment/condition, as well as contextual modifiers like abbreviations or implicit negations ("diagnosis was ruled out"). These nuances complicate creating accurate embedded representations for queries and passages. The semantic and lexical gaps limit the effectiveness of sparse models like BM25 and TF-IDF, which operate on keyword matching, leading to a shift towards dense models that can automatically learn and address these complexities Sivarajkumar et al. (2024).

## 2.2 DENSE RETRIEVAL MODELS

To address the failures of sparse retrieval, recent work has leveraged dense retrieval models built upon bidirectional encoders, such as BERT (Devlin et al. (2018)), to generate dense vectors for documents and queries, enabling the measurement of semantic similarity (Sawarkar et al. (2024)). Two main architectures have emerged from this paradigm. Bi-encoders, such as SBERT (Reimers & Gurevych (2019)) or DPR, encode queries and documents independently, enabling scalable and efficient similarity scoring via functions like dot product or cosine similarity. In contrast, cross-encoders, such as MonoBERT (Nogueira et al. (2019)), concatenate the query and document into a single input, allowing for deep token-level interactions that yield superior accuracy but often at a prohibitive computational cost, making them primarily suitable for re-ranking tasks.

In the medical field, BioBERT (Lee et al. (2019)) and PubMedBERT (Gu et al. (2020)) are leading domain-specific language models based on the BERT architecture. BioBERT is pre-trained on a large biomedical literature corpus (18 billion words from PubMed Abstracts and PMC full-text articles). PubMedBERT is trained from scratch on a biomedical domain corpus (PubMed) and defines its own vocabulary, thereby mitigating the limitations of the generic language captured by the BERT-base model. While these models possess deep domain knowledge, their effectiveness as bi-encoders is constrained by two fundamental limitations: the standard 512-token context window, which forces truncation of long clinical documents, and the representation bottleneck, where compressing a document's meaning into a single vector inevitably loses granular semantic nuance.

To address the restricted context window in a computationally efficient manner, ModerBERT (Warner et al. (2024)) emerges as an updated BERT-based architecture. It is specifically designed to capture a broader context (up to 8,192 tokens) more effectively than its predecessors. While the representation bottleneck persists as an inherent limitation of the bi-encoder architecture, our proposed two-stage architecture (retrieve and re-rank) aims to overcome both the context length and the representation bottleneck problems.

## 2.3 RE-RANKING ARCHITECTURES

To enable a more granular semantic capture without compromising the speed of the RAG system, the use of two-stage pipelines has been widely adopted. In this paradigm, a fast but less precise model retrieves candidate documents, which are subsequently processed by a more accurate, yet computationally expensive, model to be re-ordered, yielding documents with a higher degree of semantic alignment (de Souza P. Moreira et al. (2024), Upadhyay & Viviani (2025)). Re-rankers are commonly implemented using cross-encoder models, which achieve high accuracy but suffer from high computational latency, making them impractical for real-time applications with large candidate sets (Petrov et al. (2024), Reimers & Gurevych (2019)). Our work contributes to this line of research by proposing the use of ModernBERT, a state-of-the-art and scalable bidirectional encoder, with ColBERT (Santhanam et al. (2021)), a late-interaction model, as an efficient and precise re-ranker. For our use case, ColBERT achieves fine-grained token-level relevance scoring comparable to the accuracy of a full cross-encoder, while being significantly more efficient. This combination enables us to develop a high-performance clinical RAG system that effectively balances accuracy and computational efficiency.

## 3 METHODOLOGY

Our work introduces a modular, two-stage retrieval architecture designed to meet the demands of the clinical domain for RAG systems. In line with the "retrieve and re-rank" paradigm, we combine an efficient and fast bidirectional encoder for general initial retrieval with a high-precision, late-interaction model for fine-grained re-ranking. We fine-tune this entire pipeline end-to-end to specialise the components for the knowledge-intensive task of biomedical Question Answering (QA). This integrated approach aims to strike an effective balance between speed and semantic precision, two critical factors in real-world medical applications where both scalability and trustworthiness accuracy are essential.

### 3.1 ARCHITECTURE

Our proposed architecture is a two-stage "retrieve and re-ran" pipeline designed to balance retrieval speed with semantic precision.

The *first stage* features a **ModernBERT** model trained as a siamese bi-encoder, where both query and documents are encoded independently using the same encoder model to prioritise efficiency and scalability. Leveraging its long context window (up to 8,192 tokens), it transforms passages into 768-dimensional dense embeddings, enabling efficient indexing and rapid search across extensive clinical knowledge bases (Warner et al. (2024)). In an offline process, we pre-compute document embeddings from the knowledge base and index them in the Qdrant vector database. This offline indexing represents a one-time cost; however, ModerBERT's lightweight design enables efficient processing of large corpora and supports frequent knowledge base updates. At inference time, the model encodes the user's query into a vector, which is then used to perform a fast Approximate Nearest Neighbour (ANN) search to retrieve the top $k_{init}$ candidates. The primary function of this stage is to rapidly reduce the search space of size $N$, prioritising high recall to ensure $k_{init}$ relevant documents are passed to the next stage (with $k_{init} \ll N$).

The *second stage* employs a **ColBERTv2** re-ranker to address the representation bottleneck of the bi-encoder. Unlike bi-encoders, ColBERT does not collapse the query and document into a single vector. Instead, it employs a late-interaction mechanism to produce a contextualised embedding for each token in the query-document pair, preserving high semantic granularity. The document's relevance score is computed by comparing each token embedding from the query with all token embeddings from the document via a max-similarity (MaxSim) operation. Through this operation, the most similar document token is identified, and the final score is computed as the sum of these maximum similarities over all query tokens (Khattab & Zaharia (2020)). This token-level comparison allows for a much more granular and context-sensitive relevance assessment, while remaining computationally feasible since it only operates on the small subset of $k_{init}$ candidate passages obtained during the previous stage.

This token-level architecture is crucial for mitigating the severe lexical and semantic gaps existing in the clinical domain, as established in subsection 2.1. For instance, consider a query for "treatments for myocardial infarction" and a retrieved passage stating "tests were negative, and myocardial infarction was ruled out". A bi-encoder is likely to assign a high similarity score to this pair, as the negating context of the phrase "ruled out" is diluted during the averaging and compression stages into a single embedding. In contrast, by comparing each contextualised embedding of the query and the passage, ColBERT can correctly capture the semantic negating context of "myocardial infarction" altered by the modifier "ruled out", leading to a lower relevance score and penalising the overall document similarity score. This ability to capture fine-grained semantics nuances allows our re-ranker to discern truly relevant passages from those that the first stage may have selected based on entire document vector comparison.

The top $k$ passages obtained by the re-ranker are prepended to the original query $q$ to serve as the evidentiary basis for the generator LLM, which then synthesises the final response. The end-to-end data flow is shown in Figure 1.

For our implementation setting, we use Qdrant, an efficient, open-source vector database, to index our dense vectors. Llama 3.3 8B, a widely used open-source LLM, serves as the "generator". To isolate retrieval and re-ranking effects and ensure fair comparisons across experiments, we employ

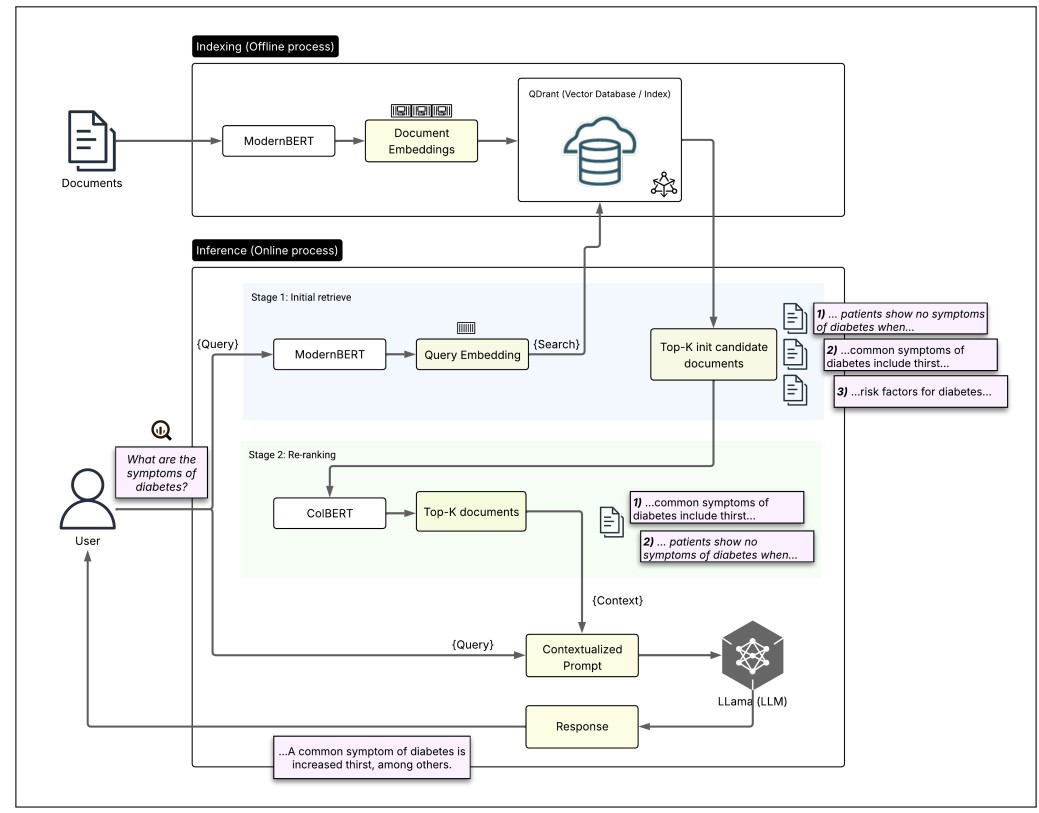

Figure 1: Diagram of the two-stage architecture with ModernBERT as the initial retriever and Col-BERT as re-ranker.

the same generator for all retrieval strategies; assessing alternative LLMs lies outside the scope of this study.

## 3.2 EXPERIMENTAL SETUP

### 3.2.1 DATASETS

The experiments were conducted using three disjoint subset derived from publicly available biomedical corpora.

- **Knowledge Base Corpus:** A sample of 1,000,000 (approx. 5%) documents from MedRAG/PubMed (Xiong et al. (2024)), comprising snippets from the PubMed corpus for specific use within the MIRAGE benchmark.

- **Pre-training dataset:** a separate, disjoint subset of 10,000 passages from MedRAG/PubMed. For this phase, we use the title–abstract pair of each document, treating the title as a proxy query and the abstract as its positive passage.

- **Fine-tuning dataset:** a sample of 10,000 question-passage pairs from PubMed_QA dataset of the BigBio project (Jin et al. (2019)), which is a structured version of PubMedQA, where each question is associated with a set of relevant passages.

### 3.2.2 BASELINES

We compare our proposed model against three competitive baselines to contextualise its performance:

- **BM25:** A robust, sparse retrieval model that represents the classic keyword-based approach.

- **DPR:** A well-established bi-encoder that serves as a powerful dense retrieval baseline.

- **MedCPT Retriever:** A zero-shot Contrastively Pre-trained Transformer model trained with real user query logs on PubMed (Jin et al. (2023)). We use the retriever-only model (passage encoder and query encoder).

- **MedGemma:** *Gemma 3* variant optimised for medical text. We use the last hidden layer of the model to obtain the passage embeddings in an encoder-only configuration (Sellergren et al. (2025)).

### 3.3 TRAINING RECIPE

Our training strategy consists of a two-phase sequential process designed to specialise each component for its specific role. First, we fine-tune the ModernBERT model for the initial retrieval task. Second, we fine-tune the ColBERT re-ranker to optimise its ability to re-sort the outputs of the first stage. The training objective for both is to learn representations that project passages semantically related to a query close to each other in the latent vector space, while pushing apart dissimilar passages. As part of our ablation studies, we experiment with two main variables during this process.

**Negative Sampling Strategies:**

*Hard Negative Mining with Random Sampling:* As a baseline, for each query-positive passage pair, we select negative passages uniformly at random from the entire corpus, discarding the gold passage and duplicates.

*Hard Negative Mining with BM25:* For each query-positive passage, we retrieve the top 42 passages with BM25 and select the 32 bottom ranks as negatives, after excluding the gold passage. This heuristic aims to provide samples with high lexical overlap but lower semantic relevance to the query. The number of negatives was standardised at 32 to match the maximum batch size that our computational resources consistently handle across all models, and we retrieve 42 candidates to ensure a sufficient pool of negatives after excluding the top-k passages used for evaluation.

*In-Batch Negative Sampling (IBNS):* Positive passages from other queries in the same training batch serve as negatives, improving computational efficiency by avoiding explicit corpus-wide searches. This strategy is used to train the first-stage ModernBERT retriever. Due to implementation constraints, the corresponding ColBERT re-ranker was instead trained using hard negatives mined from the top-ranked predictions of its corresponding fine-tuned ModernBERT retriever. This hybrid approach maintains a congruent pipeline by specifically training the re-ranker to correct the retriever errors.

**Similarity Functions:**

*Dot Product:* Computes the projection of one vector onto another, making the score sensitive to both angle and magnitude. High values may arise from large magnitudes even if vectors are not well aligned.

*Cosine Similarity:* Normalizes vectors by their L2-norm, depending only on their orientation. This yields a magnitude-invariant measure of semantic similarity, reducing bias from factors such as document length.

#### 3.3.1 EVALUATION METRICS

**End-to-End RAG.** Given our goal of enhancing medical RAG through an improved retriever, we conducted a comprehensive end-to-end evaluation on the MIRAGE benchmark (Xiong et al. (2024)), which is explicitly designed to test factuality and retrieval-grounded performance in medical QA. Its five curated datasets span complementary challenges: MMLU-Med (exam-style conceptual questions), MedQA-US (complex patient case scenarios), MedMCQA (short factual recall), PubMedQA* (question based on scientific literature), and BioASQ-Yes/No (binary literature questions). The multiple-choice and binary formats enable an objective and reproducible assessment of factual accuracy across the full RAG pipeline. For all sub-tasks, we report Accuracy, defined as the percentage of questions answered correctly.

**Retrieval Performance (Recall@k):** To directly evaluate the effectiveness of our two-stage retrieval module, we report Recall@k on an independent test set. Recall@k is defined as the pro-

| Retriever | MMLU-Med | MedQA | MedMCQA | PubMedQA* | BioASQ-Y/N | Avg. |
|---|---|---|---|---|---|---|
| BM25 | 0.5702 | 0.4705 | 0.3954 | 0.2840 | 0.4644 | 0.4369 |
| DPR | 0.5445 | **0.4902** | 0.3739 | **0.2900** | 0.3884 | 0.4174 |
| MedRAG (MedCPT) | 0.5684 | 0.4650 | 0.3964 | 0.2540 | **0.5340** | 0.4436 |
| MedGemma | **0.5803** | 0.4839 | 0.4109 | 0.2380 | 0.4725 | 0.4371 |
| M+C (CosIbns) | 0.5629 | 0.4855 | **0.4172** | 0.2780 | 0.4806 | **0.4448** |

Table 1: RAG Accuracy on the MIRAGE Benchmark. The best performing model in each category is highlighted in bold. The reported M+C (ModernBERT + ColBERT) retriever corresponds to the fine-tuned version using In-Batch Negative Sampling and Cosine Similarity (CosIbns).

portion of queries for which at least one gold-standard relevant document appears within the top $k$ retrieved passages. Achieving high Recall@k is a fundamental requirement for any RAG system, since the generator cannot produce a correct answer from evidence that was never retrieved. We report this for k={3, 5, 10}.

**Computational Efficiency:** To assess the computational performance and deployment feasibility of each model, we report the average execution time required to generate embeddings for passages (during indexing) and queries (during inference).

## 4 RESULTS

Table 1 reports the accuracy on the MIRAGE benchmark for the comparative models and for the best configuration of our proposed architecture. Our experiments were conducted with a $k = 5$ and $k_{init} = 20$ for initial retrieval in re-ranking architectures. The results for the ModernBERT + Col-BERT retriever correspond to the CosIbns setting, i.e., a model fine-tuned with Cosine Similarity as the distance metric and In-Batch Negative Sampling (IBNS) for negative passage mining following the procedure described in section 3.3.

The results show that no single configuration consistently outperforms all datasets across the five MIRAGE subtasks. Nevertheless our proposed fine-tuned architecture is not only highly competitive but also achieves the highest macro-average accuracy. With an average accuracy of 0.4448, our retriever slightly outperforms the strong MedRAG (MedCPT) baseline, which achieved an average accuracy of 0.4436. Per dataset, the top performers are: **MMLU-Med**, MedGemma with accuracy of 0.5803 ($\pm$ 0.01); **MedQA-US**, DPR with accuracy of 0.5020 ($\pm$ 0.01); **MedMCQA**, Modern-BERT + ColBERT (CosIbns) with accuracy of 0.4172 ($\pm$ 0.01); **PubMedQA***, DPR with accuracy of 0.2900 ($\pm$ 0.02); **BioASQ-Yes/No**, MedCPT with accuracy of 0.5340 ($\pm$ 0.02).

From ablation study, Table 2 reports the average accuracy on MIRAGE tasks for models trained with different negative sampling strategies and similarity functions, comparing the ModernBERT + ColBERT architecture (M+C) against the ModernBERT baseline (M). Under the M+C configuration, average accuracy improved by +3.13% with cosine similarity and +2.31% with dot product. In contrast, BM25 and Random Sampling strategies yielded marginal gains ($<$+0.3%) and even negative effects under cosine similarity. Overall, these results highlight that performance is highly dependent on the negative sampling strategy. When combined with IBNS, cosine similarity consistently emerges as the most effective distance metric in this test: our best-performing model, M+C CosIbns, reached an average accuracy of 0.4448, outperforming its dot product counterpart (0.4367). This validates the combination of cosine similarity and IBNS as the optimal fine-tuning strategy in our experiments.

While end-to-end accuracy defines the utility of a RAG system, direct evaluation of the retriever module is crucial for understanding its core effectiveness. On a 10k PubMedQA subset, Recall@k results (Figure 2) reveal three insights: (i) fine-tuning is crucial, as zero-shot ModernBERT achieves only 37.9% Recall@10, while our best configuration (DotIbns) reaches 92.8%; (ii) training methodology matters, since suboptimal strategies, such as CosRand, collapse performance to 1.7%, underscoring the importance of IBNS; and (iii) the two-stage pipeline adds further value, with ColBERT re-ranking improving Recall@10 from 92.8% to 93.8%, making our retriever directly competitive with the MedCPT baseline (96.7%).

| Sampling strategy | Cosine similarity | | Dot product | |
|---|---|---|---|---|
| | Avg. Acc | Δ vs (base) | Avg. Acc | Δ vs (base) |
| Rand | 0.4029 | -1.06% | 0.4159 | 0.24% |
| BM25 | 0.4083 | -0.52% | 0.4164 | 0.29% |
| IBNS | 0.4448 | **3.13%** | 0.4367 | **2.31%** |

Table 2: Average accuracy and Δ relative to the not fine-tuned model for different negative sampling strategies in re-ranker configuration (M+C).

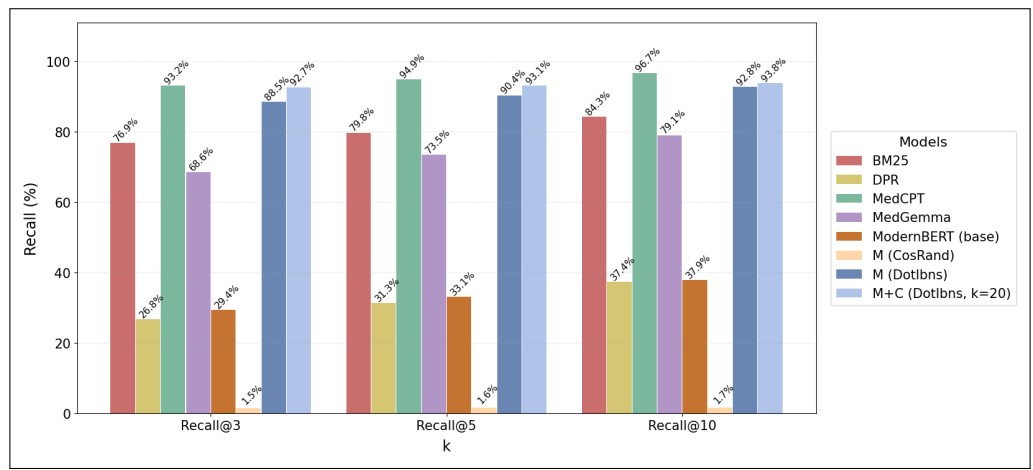

Figure 2: Recall@k obtained by base and fine-tuned models in their corresponding configurations with re-ranker (M+C) and without re-ranker (M).

Table 3 shows the average passage encoding time during batch indexing (500 passages per batch), as well as the average query encoding time and the re-ranking stage during inference. Our ModernBERT-based retriever encodes passages in 0.80 ms—over 7.5× faster than MedCPT and 49× faster than MedGemma—making it particularly suited for dynamic knowledge bases that require frequent updates. At inference, the two-stage pipeline introduces extra latency (57.66 ms vs. 26.53 ms for MedCPT) but remains competitive for real-time applications. Overall, the system balances acceptable query latency with a substantial advantage in indexing efficiency, ensuring practicality for clinical deployment.

| Model | Index (ms/passage) | Query (ms) | Re-rank (ms) | Total Inference (ms) |
|---|---|---|---|---|
| BM25 | 0.05 | 3,922.50 | - | 3,922.50 |
| DPR | 12.90 | 20.37 | - | 20.37 |
| MedCPT | 6.16 | 26.53 | - | 26.53 |
| MedGemma | 39.31 | 83.32 | - | 83.32 |
| M (CosIbns) | **0.80** | 41.84 | - | 41.84 |
| M+C (CosIbns) | **0.80** | 31.40 | 26.26 | 57.66 |

Table 3: Indexing and Inference Latency (in milliseconds). The indexing process was performed using a batch size of 500 passages.

Our analysis shows that our two-stage retriever (ModernBERT + ColBERT) achieves competitive performance on the MIRAGE benchmark, outperforming state-of-the-art models like MedCPT and MedGemma. The results yield two critical insights for building high-performance, specialised RAG systems.

First, a congruent end-to-end fine-tuning strategy is essential, as the value of the ColBERT re-ranker is highly dependent on its alignment with the first-stage retriever. Our findings show that training

the re-ranker on hard negatives identified by the retriever is a crucial step that aligns their latent spaces and significantly boosts performance, whereas training the components independently offers marginal or even detrimental effects.

Second, performance is highly sensitive to alignment between the training data distribution and the target task. Our system excels at factual-recall tasks (MedMCQA) that mirror our training data. However, its performance is inconsistent on tasks with a significant distribution shift from the training data, such as those worded to engage in clinical reasoning (MMLU-Med) or those with substantially different query structures (e.g., long-token questions).

Finally, our architecture proves to be a practical and scalable solution suitable for clinical decision support or research query systems. While the re-ranking stage increases the total query latency compared to a standalone bi-encoder, the total inference time remains well within the threshold for interactive applications (Nielsen (1994)). More importantly, the exceptional indexing speed of our ModernBERT retriever makes the system highly maintainable, especially in clinical environments where knowledge bases are dynamic and may require frequent updates as new evidence becomes available (e.g. through new clinical trials).

## 5    CONCLUSION

In this work, we tackled the challenges of retrieval in biomedical RAG systems by proposing and evaluating a two-stage architecture combining an efficient ModernBERT retriever with a precise late-interaction ColBERT re-ranker. Our comprehensive experiments on the MIRAGE benchmark yield two critical findings: first, a cohesive end-to-end fine-tuning strategy is crucial for achieving high performance. When the ColBERT re-ranker is specifically fine-tuned using hard negatives mined from the trained ModernBERT retriever, the solution can outperform leading models, such as MedCPT and MedGemma. Second, this staged training is necessary to unlock the full potential of the two-stage pipeline, as only an aligned retriever and re-ranker can deliver substantial performance gains. Our optimally tuned system achieves state-of-the-art average accuracy (0.4448) on MIRAGE and demonstrates a practical advantage with indexing speeds over 7.5x faster than baselines. Ultimately, we present a robust and efficient framework that balances accuracy and scalability, providing a practical pathway for deploying trustworthy and maintainable RAG systems in the dynamic clinical domain.

### 5.1    LIMITATIONS AND FUTURE WORK

While our results are promising, we acknowledge limitations within our study: the proposed two-stage architecture's performance is fundamentally capped by the recall of the first-stage retriever; if the proper evidence is not present within the initial candidate set, even a perfect re-ranker cannot recover it. Future work should therefore focus on mitigating this dependency. We identify two promising directions for future work: **Optimisation of retrieval parameters**. Our experiments used a fixed candidate set size ($k_{\text{init}}$), but a systematic exploration is needed to understand its impact on recall and latency. Furthermore, scaling our experiments to a larger representative sample of the PubMed corpus is necessary for a conclusive comparison with state-of-the-art models. **Architectural development**. In this work, our models were trained in a siamese configuration, meaning that the same encoder was used for both queries and passages. We plan to explore an asymmetric (dual-encoder) architecture for the ModernBERT retriever, as employed by leading models like MedCPT. Training separate encoders for queries and passages could enhance retrieval accuracy, thereby providing a higher-quality candidate set for the re-ranker and improving the overall pipeline performance.

## 6    USE OF LARGE LANGUAGE MODELS (LLMS)

For this work, an LLM (GPT-5) was used solely for writing assistance, specifically to polish grammar, style, and clarity and to condense the main text. No LLMs were used for research ideation or methodological development. The authors take full responsibility for the content of this paper.

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

## A  PROMPT TEMPLATE

```
<|begin_of_text|><|start_header_id|>system<|end_header_id|>
You are a meticulous and highly accurate medical AI assistant.
Your sole purpose is to analyze provided medical documents
to answer a multiple-choice question.
You MUST strictly follow all instructions and provide your
output ONLY in the specified JSON format.
<|eot_id|><|start_header_id|>user<|end_header_id|>
### TASK ###
Analyze the documents and answer the question according to
the rules below.

### CONTEXT DOCUMENTS ###
{context_str}

### QUESTION AND OPTIONS ###
**Question:** {query}

**Options:**
{options}

### OUTPUT RULES ###
1.  **Reasoning:** First, think step-by-step to arrive at
your conclusion. Your entire thought process must be captured
```

```
in the `step_by_step_thinking` field.
2.  **Relevance Check:** Determine if the context documents
were relevant and necessary to answer the question. Use "YES"
or "NOT" for the `relevant_context` field.
3.  **Final Answer:** Choose one single, definitive letter
corresponding to the correct option. This will be the value
for the `answer_choice` field.
4.  **Strict JSON Format:** Your entire response MUST be a
single, raw JSON object. Do not write any text, explanation,
or markdown formatting (like ```json) before or after the
JSON object.

Your response must conform to this exact JSON structure:
```json
{{
  "step_by_step_thinking": "Your detailed analysis and
  reasoning to reach the answer.",
  "relevant_context": "YES",
  "answer_choice": "C"
}}
<|eot_id|><|start_header_id|>assistant<|end_header_id|>
```

# B  DETAILED RESULTS

## B.1  ACCURACY MIRAGE BENCHMARK

| Model | MMLU-Med | MedQA | MedMCQA | PubMedQA* | BioASQ-Y/N | Avg. |
|---|---|---|---|---|---|---|
| BM25 | 57.02 ±1.50 | 47.05 ±1.40 | 39.54 ±0.76 | 28.40 ±2.02 | 46.44 ±2.01 | 43.69 |
| DPR | 54.45 ±1.51 | 49.02 ±1.40 | 37.39 ±0.75 | 29.00 ±2.03 | 38.84 ±1.96 | 41.74 |
| MedRAG (MedCPT) | 56.84 ±1.50 | 46.50 ±1.40 | 39.64 ±0.76 | 25.40 ±1.95 | **53.40** ±2.01 | 44.36 |
| MedGemma | 58.03 ±1.50 | 48.39 ±1.40 | 41.09 ±0.76 | 23.80 ±1.91 | 47.25 ±2.01 | 43.71 |
| ModernBERT (base) | 54.36 ±1.51 | 48.39 ±1.40 | 37.39 ±0.75 | 21.00 ±1.82 | 40.13 ±1.97 | 40.25 |
| ModernBERT (CosBM25) | 56.93 ±1.50 | 47.37 ±1.40 | 38.13 ±0.75 | 24.00 ±1.91 | 38.19 ±1.95 | 40.92 |
| ModernBERT (CosIbns) | **58.31** ±1.49 | 45.80 ±1.40 | 40.16 ±0.76 | 27.00 ±1.98 | 46.93 ±2.01 | 43.64 |
| ModernBERT (CosRand) | 56.11 ±1.50 | 48.86 ±1.40 | 38.56 ±0.75 | 18.80 ±1.75 | 40.78 ±1.98 | 40.62 |
| ModernBERT (DotBM25) | 56.01 ±1.50 | 49.57 ±1.40 | 38.61 ±0.75 | 24.20 ±1.92 | 39.32 ±1.97 | 41.54 |
| ModernBERT (DotIbns) | 56.38 ±1.50 | 48.39 ±1.40 | 40.21 ±0.76 | 25.40 ±1.95 | 48.71 ±2.01 | 43.82 |
| ModernBERT (DotRand) | 52.25 ±1.51 | **50.20** ±1.40 | 37.80 ±0.75 | **29.20** ±2.03 | 37.06 ±1.94 | 41.30 |
| ModernBERT + ColBERT (base) | 54.64 ±1.51 | 48.47 ±1.40 | 39.35 ±0.76 | 23.20 ±1.89 | 41.10 ±1.98 | 41.35 |
| ModernBERT + ColBERT (CosBM25) | 56.01 ±1.50 | 47.92 ±1.40 | 38.13 ±0.75 | 23.60 ±1.90 | 38.51 ±1.96 | 40.83 |
| ModernBERT + ColBERT (CosIbns) | 56.29 ±1.50 | 48.55 ±1.40 | **41.72** ±0.76 | 27.80 ±2.00 | 48.06 ±2.01 | **44.48** |
| ModernBERT + ColBERT (CosRand) | 55.28 ±1.51 | 49.73 ±1.40 | 36.55 ±0.74 | 20.40 ±1.80 | 39.48 ±1.97 | 40.29 |
| ModernBERT + ColBERT (DotBM25) | 55.37 ±1.51 | 48.55 ±1.40 | 39.18 ±0.76 | 25.60 ±1.95 | 39.48 ±1.97 | 41.64 |
| ModernBERT + ColBERT (DotIbns) | 56.11 ±1.50 | 48.63 ±1.40 | 39.71 ±0.76 | 26.80 ±1.98 | 47.09 ±2.01 | 43.67 |
| ModernBERT + ColBERT (DotRand) | 55.65 ±1.51 | 49.65 ±1.40 | 36.98 ±0.75 | 28.60 ±2.02 | 37.06 ±1.94 | 41.59 |

Table 4: Accuracy (%) obtained in MIRAGE benchmark task for different retriever configurations. The highest accuracy is in bold. (Cos-: Cosine similarity, Dot-: Dor product; -BM25: BM25 sampling, -Rand: Random sampling, -Ibns: In-Batch Negative sampling)

## B.2 RECALL@K

| Model Configuration | Recall@3 | Recall@5 | Recall@10 |
|---|---|---|---|
| *Baselines* | | | |
| BM25 | 0.769 | 0.798 | 0.843 |
| DPR | 0.268 | 0.313 | 0.374 |
| MedCPT | 0.932 | 0.949 | **0.967** |
| MedGemma | 0.686 | 0.7351 | 0.790 |
| *Base Models M+C (Zero-shot)* | | | |
| ModernBERT (base) | 0.294 | 0.331 | 0.379 |
| *Our Models (Fine-tuned)* | | | |
| M (CosBM25) | 0.014 | 0.016 | 0.021 |
| M (CosIbns) | 0.856 | 0.884 | 0.906 |
| M (CosRand) | 0.015 | 0.016 | 0.017 |
| M (DotBM25) | 0.002 | 0.003 | 0.010 |
| M (DotIbns) | 0.885 | 0.904 | 0.928 |
| M (DotRand) | 0.000 | 0.000 | 0.001 |
| M + C (base) ($k_{init} = 20$) | 0.434 | 0.435 | 0.436 |
| M + C (CosBM25) ($k_{init} = 20$) | 0.026 | 0.026 | 0.026 |
| M + C (CosIbns) ($k_{init} = 20$) | 0.914 | 0.922 | 0.927 |
| M + C (CosRand) ($k_{init} = 20$) | 0.019 | 0.019 | 0.019 |
| M + C (DotBM25) ($k_{init} = 20$) | 0.011 | 0.011 | 0.011 |
| M + C (DotIbns) ($k_{init} = 20$) | 0.927 | 0.931 | 0.938 |
| M + C (DotRand) ($k_{init} = 20$) | 0.004 | 0.004 | 0.004 |
| M + C (base) ($k_{init} = 100$) | 0.581 | 0.586 | 0.594 |
| M + C (CosBM25) ($k_{init} = 100$) | 0.053 | 0.053 | 0.053 |
| M + C (CosIbns) ($k_{init} = 100$) | 0.941 | 0.949 | 0.955 |
| M + C (CosRand) ($k_{init} = 100$) | 0.043 | 0.043 | 0.043 |
| M + C (DotBM25) ($k_{init} = 100$) | 0.032 | 0.032 | 0.032 |
| M + C (DotIbns) ($k_{init} = 100$) | **0.945** | **0.954** | 0.962 |
| M + C (DotRand) ($k_{init} = 100$) | 0.029 | 0.029 | 0.029 |

Table 5: Retriever Performance: Recall@k on PubMedQA dataset test. M (ModernBERT), C (Col-BERT), $k_{init}$ defines the $k$ initial retrieval.

