# OpenReview forum: "ModernBERT + ColBERT: Enhancing biomedical RAG through an advanced re-ranking retriever"
_ICLR.cc/2026/Conference — Submitted to ICLR 2026_

### Official Review · Reviewer_qGRQ · 2025-10-30

**Soundness:** 1
**Presentation:** 2
**Contribution:** 2
**Rating:** 4
**Confidence:** 4

**Summary:**

The paper presents a two-stage retrieval pipeline for biomedical RAG, comprising a lightweight ModernBERT bi-encoder for efficient initial retrieval and a ColBERTv2 late-interaction model for fine-grained re-ranking. The system is trained end-to-end on 10k QA pairs from PubMedQA and evaluated on the MIRAGE benchmark. Experiments demonstrate that the proposed method improves both retrieval quality and downstream RAG performance. The authors further highlight the benefits of joint fine-tuning and the efficiency of their parameter-efficient design.

**Strengths:**

- The combination of ModernBERT and ColBERTv2 is well-motivated and conceptually sound for clinical information retrieval.
- The experiments demonstrate the competitive accuracy of the proposed method, with the best average accuracy on the MIRAGE benchmark.
- The proposed system is practically efficient, making it useful for continuously updated biomedical knowledge bases.

**Weaknesses:**

- The improvement over MedCPT (0.4448 vs. 0.4436 on MIRAGE) appears marginal. It is unclear if the gain is statistically significant.
- The choice of ModernBERT as the bi-encoder is not fully justified. It is unclear whether it outperforms established domain-specific encoders such as BioBERT or PubMedBERT for the given tasks.
- The rationale for using ColBERTv2 as the re-ranker needs more explanation. Alternative approaches, such as the cross-encoder reranker in MedCPT, are not compared in the experiments.
- Details on training cost and resources (e.g., time, hardware) are limited, making it difficult to assess the efficiency relative to off-the-shelf retrievers that require no additional training.

**Questions:**

- The MIRAGE performance gain over MedCPT seems small. Is the improvement statistically significant?
- Why was ModernBERT chosen as the bi-encoder? How does it compare with BioBERT or PubMedBERT on the same benchmarks?
- What motivated the choice of ColBERTv2 as the re-ranker instead of other cross-encoder rerankers? Were any comparisons attempted?
- Can the authors provide more training cost details, such as hardware and training duration?

---

### Official Review · Reviewer_NMqc · 2025-10-31

**Soundness:** 3
**Presentation:** 3
**Contribution:** 2
**Rating:** 4
**Confidence:** 3

**Summary:**

This paper proposes a two-stage retrieval module for biomedical RAG that combines a lightweight ModernBERT bi-encoder (fast initial retrieval) with a ColBERTv2 late-interaction re-ranker (token-level precision). The authors fine-tune both stages (with several negative-sampling strategies and similarity functions) on PubMed/PubMedQA subsets, evaluate retrieval Recall@k and end-to-end RAG accuracy on the MIRAGE benchmark, and report that the ModernBERT+ColBERT pipeline (with in-batch negatives + cosine similarity) achieves the highest macro-average MIRAGE accuracy. They provide ablations showing the re-ranker improves Recall@k only when jointly fine-tuned with the retriever, and report latency/indexing numbers and dataset / training setup details.

**Strengths:**

- The paper evaluates multiple negative sampling strategies, similarity functions, and reports retrieval (Recall@k), end-to-end RAG accuracy across 5 MIRAGE sub-tasks, plus latency/indexing tradeoffs. The ablation showing re-ranker usefulness only when jointly fine-tuned is valuable.
- The strong emphasis on indexing speed and realistic deployment tradeoffs (indexing time, inference latency) is useful for practitioners working with dynamic biomedical corpora.

**Weaknesses:**

- Marginal end-to-end gain, statistical significance unclear. The reported macro-average improvement over MedCPT is very small (0.4448 vs 0.4436).
- Evaluation limited to one generator LLM. All RAG evaluations use a single generator (Llama-3.3 8B). While the retrieval module is the focal point, end-to-end RAG accuracy depends heavily on the generator and prompt template.

**Questions:**

See Weaknesses.

---

### Official Review · Reviewer_sB88 · 2025-11-01

**Soundness:** 2
**Presentation:** 3
**Contribution:** 3
**Rating:** 4
**Confidence:** 2

**Summary:**

This paper proposes a 2stage retrieval (ModernBERT+ColBERT) to improve biomedical RAG performance. The model is jointly fine-tuned on PubMedQA and evaluated on the MIRAGE and achiev avg accuracy (0.4448) while being faster in indexing than MedCPT. Results highlight that joint retriever–re-ranker fine-tuning is essential for optimal performance.

**Strengths:**

The proposed techinique (ModernBERT+ ColBERT) of using  two-stage retrieval architecture is technically smart. Combining ModernBERT’s long-context bi-encoder with ColBERT’s late-interaction mechanism leverages both scalability and semantic precision. Also the results are presented with clarity and transparency such as including ablation studies that isolate the effects of similarity metrics, sampling strategies, and fine-tuning alignment

**Weaknesses:**

The MIRAGE benchmark for biomedical QA is a great starting point. however, it also means that the current experiment setup of this paper is restricted to just one domain. The paper would benefit from cross-domain/corpus validation/testing to substantiate claims of generality and scalability across different biomedical text types.

I found few grammatical errors (minor ones). for examp: model name “ModernBERT” is misspelled as “ModerBERT” in line L147. Just wanted to bring attention in case missed.

I believe that the paper can also benefit if little qualitative insight was also included. that would tell why certain retrieval or re-ranking errors occur. Examples of retrieved passages and failure cases could help elucidate the system’s limitations .

**Questions:**

1. This is related to the weakness section but i am curious to know if there was any additional experiments to see how the model perform on other biomedical text types such as clinical notes, EHR summaries? If it was a different domain, would that significantly impact retrieval quality?

2. Would it be possible to add a qualitative analyhsis of the results so that the readers can understand common eror points?

---

### Official Review · Reviewer_5pjf · 2025-11-01

**Soundness:** 2
**Presentation:** 2
**Contribution:** 1
**Rating:** 2
**Confidence:** 4

**Summary:**

This paper proposes a two-stage retrieval architecture for biomedical Retrieval-Augmented Generation (RAG), combining a ModernBERT bi-encoder for initial dense retrieval with a ColBERTv2 late-interaction model for re-ranking. The system is fine-tuned using PubMedQA question–passage pairs and evaluated on the MIRAGE benchmark. Results show that the ColBERT re-ranking improves retrieval Recall@k and leads to state-of-the-art average RAG accuracy (0.4448), slightly outperforming strong baselines such as MedCPT, while maintaining significantly faster indexing speed (over 7.5× improvement). The paper claims to offer a practical balance between efficiency and accuracy for biomedical QA.

**Strengths:**

- The proposed method achieves state-of-the-art performance while being computationally efficient.

**Weaknesses:**

- The performance is not strong enough. In table 1, the proposed method only outperforms baselines in MedMCQA and in average. However, the average accuracy is not that high when compared with MedCPT (0.4448 vs 0.4436). The recall@k in Figure 2 also shows that the proposed method doesn't outperform MedCPT.
- The innovation is also limited. As the title suggests, the proposed method basically combines ModernBERT for retrieval and ColBERT for re-ranking.
- The index time is largely reduced. However, given the time for reranking, the total time is longer than MedCPT.

**Questions:**

- Which model do you use for DPR? I don't think this is specified in the paper.
- Why the recall@k of MedCPT is the highest, but the scores in table 1 are not always the highest?

---

### Meta-Review · Area_Chair_oQEM · 2026-01-06

**Summary:**

We have four reviewers who have carefully checked this paper and they all lean towards rejection. The primary concerns include both the novelty of the proposed method and the limited performance. No rebuttal was provided. Thus, I recommend that the paper should be revised and resubmitted with more comprehensive revision.

**Reviewer Concerns:**

The concerns include both the novelty and the limited performance, no rebuttal was provided. Thus, the concerns still outstanding.

**Reviewer Scores:**

No rebuttal was provided.

---

### Decision · Program_Chairs · 2026-01-26

Reject